# Counterfactual Temporal Point Processes

**Kimia Noorbakhsh**[*]
Sharif University of Technology
kimianoorbakhsh@gmail.com

**Manuel Gomez Rodriguez**
Max Planck Institute for Software Systems
manuelgr@mpi-sws.org

## Abstract

Machine learning models based on temporal point processes are the state of the art in a wide variety of applications involving discrete events in continuous time. However, these models lack the ability to answer counterfactual questions, which are increasingly relevant as these models are being used to inform targeted interventions. In this work, our goal is to fill this gap. To this end, we first develop a causal model of thinning for temporal point processes that builds upon the Gumbel-Max structural causal model. This model satisfies a desirable counterfactual monotonicity condition, which is sufficient to identify counterfactual dynamics in the process of thinning. Then, given an observed realization of a temporal point process with a given intensity function, we develop a sampling algorithm that uses the above causal model of thinning and the superposition theorem to simulate counterfactual realizations of the temporal point process under a given alternative intensity function. Simulation experiments using synthetic and real epidemiological data show that the counterfactual realizations provided by our algorithm may give valuable insights to enhance targeted interventions.

## 1 Introduction

In recent years, machine learning models based on temporal point processes have become increasingly popular for modeling discrete event data in continuous time [1, 2]. This type of data is ubiquitous in a wide range of application domains, from social and information networks to finance or epidemiology. For example, in social and information networks, events may represent users' posts, clicks or likes [3, 4]; in finance, they may represent buying and selling orders [5]; or, in epidemiology, they may represent when an individual gets infected or recovers [6]. In many of these domains, these models have become state of the art at predicting future events given a sequence of past events [7].

Building upon the above models, a recent line of work [8–11] has developed machine learning methods to automate online, adaptive targeted interventions using reinforcement learning and stochastic optimal control. While this line of work has shown early promise, particularly in personalized teaching and viral marketing, there are many high-stakes application in which targeted interventions are unlikely to be automated. For example, in epidemiology, fine-grained interventions that are targeted at particular sites or individuals (*e.g.*, hygienic measures at work places, closures of schools, or contact tracing) are likely to be decided by governments, policy makers and health authorities, at least in the foreseeable future. In this work, our goal is to develop machine learning methods that, given the outcome of an intervention implemented in the past, are able to assist decision makers at implementing better interventions in these high-stakes applications.

More specifically, we focus on facilitating counterfactual thinking, a type of thinking that has been argued to help humans correct and improve behavior that has been unsuccessful in the past [12, 13]. In counterfactual thinking, given a history of past events that have already occurred, one asks what past events would have instead occurred if certain intervention had been in place. For example, in epidemiology, assume that, during a pandemic, a government decides to implement business

---

[*]This work was done during Kimia Noorbakhsh's internship at MPI-SWS.

36th Conference on Neural Information Processing Systems (NeurIPS 2022).

restrictions every time the weekly incidence—the (relative) number of new cases—is larger than certain threshold but unfortunately the incidence nevertheless spirals out of control. In this case, counterfactual thinking would help the government understand retrospectively to what extent the incidence would have grown had a lower threshold been implemented[2].

**Our contributions.** Our starting point is Lewis' thinning algorithm [20], one of the most popular techniques for sampling in temporal point processes. Lewis' thinning algorithm first samples a sequence of potential events from a temporal point process with a constant intensity that upper bounds the intensity of the temporal point process of interest. Then, it accepts each of these events with probability proportional to the ratio between the intensity of the temporal point process of interest at the time of the sampled event and the constant intensity. In our work, the key idea is to augment the above thinning process using a particular class of structural equation models (SCMs)—the Gumbel-Max SCM [21]. This causal model satisfies a desirable monotonicity condition and, given a sequence of accepted and rejected events, it allows us to reliably estimate what events would have been accepted and rejected under an alternative intensity. Put differently, it can be used to answer counterfactual questions about a set of previously accepted and rejected events by the thinning algorithm.

Unfortunately, the above causal model on its own is not sufficient to answer counterfactual questions about observed sequence of real events. This is because, in general, real event data is not *generated* by a thinning process and, as a consequence, it does not *include* rejected events. However, we are able to overcome this limitation by using the superposition theorem [22] to sample *plausible* sequences of events that the thinning process would have rejected if it had accepted the observed sequence of real events. Then, these generated sequences of rejected events, together with the observed events, can be fed into the above causal model of thinning to sample counterfactual events given an alternative intensity. Importantly, while our causal model of thinning only allows for inhomogeneous Poisson processes (*i.e.*, it requires the intensities of interest to be deterministic), we can still use it to sample counterfactual events from linear Hawkes processes [23], a popular type of temporal point processes with stochastic self-exciting intensities, by exploiting their branching process interpretation [24]. Finally, we evaluate our sampling algorithm using both synthetic and real epidemiological data and show that the counterfactual events provided by our algorithm can give valuable insights to enhance targeted interventions[3].

**Further related work.** The literature on temporal point processes related to causal inference has mostly focused on measuring causal influence by means of, *e.g.*, Granger causality [25–28], integrated cumulants [29, 30] or Wold processes [31, 32], and on predicting quantities related to an interventional distribution of interest [33–35] (*e.g.*, conditional average treatment effect (CATE)). However, there exist a few notable exceptions, which have focused on counterfactual reasoning [36–38]. The work by Schulam and Saria [36] focuses on reasoning about the counterfactual distributions of the marks in a marked temporal point process, rather than reasoning about the counterfactual intensities of the events as we do. The works by Ryalen et al. [37] and Roysland [38] focus on survival analysis, *i.e.*, temporal point processes that terminate after one event, and are very different to ours at a technical level. In contrast, we focus on temporal point processes with multiple events[4].

More broadly, the literature on causal inference has a long and rich history [40]. However, most of this literature has used counterfactual reasoning to predict quantities related to the interventional distribution of interest such as, *e.g.*, the conditional average treatment effect (CATE). A few recent notable exceptions are by Oberst and Sontag [21] and Tsirtsis et al. [41], which have used the Gumbel-Max SCM to reason about counterfactual distributions in Markov decision processes (MDPs), and by Buesing et al. [42] and Pitis et al. [43], which have used counterfactuals to improve the training of reinforcement learning agents. However, to the best of our knowledge, the Gumbel-Max SCM has not been used previously to reason about counterfactual events in temporal point processes.

## 2 Preliminaries

In this section, we first briefly revisit the frameworks of temporal point processes [44] and structural causal models [45].

---

[2] Note that existing epidemiological models [14], including those developed in the context of COVID-19 [15–19], are unable to answer such counterfactual questions—they can only predict what the (average) future would look like under certain interventions given the past.

[3] To facilitate research in this area, we release an open-source implementation of our algorithms and data at https://github.com/Networks-Learning/counterfactual-ttp.

[4] The work by Zhang et al. [39] is contemporary to ours and presents a methods to estimate individual treatment effects (ITE) on the intensity of a temporal point process.

**Temporal point processes.** A temporal point process is a stochastic process whose realization consists of a sequence of discrete events localized in continuous time, $\mathcal{H} = \{t_i \in \mathbb{R}^+ \mid i \in \mathbb{N}^+, t_i < t_{i+1}\}$. A temporal point process can be equivalently represented as a counting process, $N(t)$, which records the number of events before time $t$. Moreover, in an infinitesimally small time window $dt$ around time $t$, it is assumed that only one event can happen, *i.e.*, $dN(t) \in \{0, 1\}$.

A temporal point process is typically characterized via its intensity function $\lambda(t) \geq 0$, which determines the probability of observing an event in $[t, t + dt)$, *i.e.*,

$$\lambda(t)dt = \mathbb{P}\{dN(t) = 1\} = \mathbb{E}[dN(t)]. \tag{1}$$

In general, the intensity $\lambda(t)$ may depend on the history $\mathcal{H}(t) = \{t_i \in \mathcal{H} \mid t_i < t\}$ up to time $t$ and its functional form is often chosen to capture the phenomena of interest. Throughout the paper, we will consider inhomogeneous Poisson processes, *i.e.*, $\lambda(t) = g(t)$, where $g(t) \geq 0$ is a time-varying function, and linear Hawkes processes, *i.e.*,

$$\lambda(t) = \mu + \alpha \sum_{t_i \in \mathcal{H}(t)} g(t - t_i), \tag{2}$$

where $\mu \geq 0$ and the second term, with $\alpha \geq 0$, $g(t) \geq 0$ and $g(t) = 0$ for all $t < 0$, denotes the influence of previous events on the current intensity[5].

**Structural causal models.** Given a set of random variables $\boldsymbol{X} = \{X_1, \ldots, X_n\}$, a structural causal model (SCM) $\mathcal{C}$ defines a complete data-generating process via a collection of assignments

$$X_i := f_i(\mathbf{PA}_i, U_i), \tag{3}$$

where $\mathbf{PA}_i \subseteq \boldsymbol{X} \backslash X_i$ are the direct causes of $X_i$, $\boldsymbol{U} = \{U_1, \ldots, U_n\}$ are noise variables, and $P(\boldsymbol{U})$ denotes the (prior) distribution of the noise variables. Here, note that, given an observational distribution $P(X_1, \ldots, X_n)$, there always exists a distribution for the noise variables and functions $f_i$ so that $P = P^{\mathcal{C}}$, where $P^{\mathcal{C}}$ is the distribution entailed by $\mathcal{C}$. In general, the noise variables $\boldsymbol{U}$ may not be jointly independent if there are unmeasured confounders[6]. However, in our work, we will assume there are no unmeasured confounders and $\boldsymbol{U}$ are jointly independent.

Given a SCM $\mathcal{C}$, we can express (atomic) interventions $\mathcal{I}$ using the *do-operator*, *e.g.*, $\mathcal{I} = \mathrm{do}(X_i = x)$ corresponds to replacing the causal mechanism $f_i(\mathbf{PA}_i, U_i)$ with $x$. The intervened SCM is typically denoted as $\mathcal{C}^{\mathcal{I}}$ and the interventional distribution entailed by the intervened SCM as $P^{\mathcal{C};\mathcal{I}}$. Moreover, given a SCM $\mathcal{C}$ and an observed realization of assignments $\boldsymbol{X} = \boldsymbol{x}$, we can define a counterfactual SCM $\mathcal{C}_{\boldsymbol{X}=\boldsymbol{x}}$ where the noise $\boldsymbol{U}$ variables are distributed according to the posterior distribution $P(\boldsymbol{U} \mid \boldsymbol{X} = \boldsymbol{x})$ and not necessarily jointly independent anymore. Counterfactual statements can now be seen as interventions in a counterfactual SCM $\mathcal{C}_{\boldsymbol{X}=\boldsymbol{x}}$ and, given an intervention $\mathcal{I}$, we denote the interventional counterfactual distribution entailed by $\mathcal{C}_{\boldsymbol{X}=\boldsymbol{x}}^{\mathcal{I}}$ as $P^{\mathcal{C}\mid\boldsymbol{X}=\boldsymbol{x};\mathcal{I}}$. However, the posterior distribution of the noise variables may be non-identifiable without further assumptions. This is because there may be several noise distributions and functions $g_i$ consistent with the observational distribution but result in different counterfactual distributions.

In the context of binary random variables, monotonicity is an assumption that avoids the above mentioned non-identifiability issues—it restricts the class of possible SCMs to those which all yield equivalent counterfactual distributions over a binary variable of interest [46, 21]. More specifically, a SCM $\mathcal{C}$ of a binary variable $Y$ is monotonic with respect to a binary variable $T$ if and only if the condition

$$P^{\mathcal{C};\mathrm{do}(T=t)}(Y = y) \geq P^{\mathcal{C};\mathrm{do}(T=t')}(Y = y)$$

implies that $P^{\mathcal{C}\mid Y=y, T=t';\mathrm{do}(T=t)}(Y = y') = 0$, where $y' \neq y$.

## 3 A Causal Model of Thinning

In this section, we first revisit Lewis' thinning algorithm [20], one of the most popular techniques to simulate event data from inhomogeneous Poisson processes. Then, we augment this classical algorithm using a particular class of SCMs satisfying the monotonicity assumption, the Gumbel-Max SCMs [21]. Finally, we demonstrate that the resulting algorithm can be used to answer counterfactual questions about a set of previously simulated events.

---

[5]The function $g(t)$ is often called triggering kernel.

[6]An unmeasured confounder is an unobserved variable that is a direct cause of two (observed) variables $X_i$ and $X_j$.

---

**Algorithm 1** It samples a counterfactual sequence of accepted events given a sequence of accepted and rejected events provided by Lewis' thinning algorithm

---
1: **Input**: $\lambda_m(t), \lambda_{m'}(t), \mathcal{H}_m, \mathcal{H}_{\max}, \lambda_{\max}$.
2: **Initialize**: $\mathcal{H}_{m'} = \emptyset$.

3: **function** ACC($\lambda_m(t), \lambda_{m'}(t), \mathcal{H}_m, \mathcal{H}_{\max}, \lambda_{\max}$)
4:     $\mathcal{H}_{m'} \leftarrow \emptyset$
5:     **for** $t_i \in \mathcal{H}_{\max}$ **do**
6:         $x_i \leftarrow \mathbf{1}[t_i \in \mathcal{H}_m]$
7:         $x_i' \sim P^{\mathcal{C} \,|\, X_i = x_i, \Lambda_i = \lambda_m(t_i)\,;\, \mathrm{do}(\Lambda_i = \lambda_{m'}(t_i))}(X)$
8:         **if** $x_i' = 1$ **then**
9:             $\mathcal{H}_{m'} \leftarrow \mathcal{H}_{m'} \cup \{t_i\}$
10:         **end if**
11:     **end for**
12:     **return** $\mathcal{H}_{m'}$
13: **end function**

---

Let $\mathcal{M}$ be a set of inhomogeneous Poisson processes of interest and, for each $m \in \mathcal{M}$, assume its corresponding intensity function $\lambda_m(t) \leq \lambda_{\max}$ for all $t \in \mathbb{R}^+$. To sample a sequence of events $\mathcal{H}_m$ from any process $m \in \mathcal{M}$, Lewis' thinning algorithm first samples a sequence of potential events $\mathcal{H}_{\max}$ from a homogenous Poisson process with intensity $\lambda_{\max}$. Then, for each event $t_i \in \mathcal{H}_{\max}$, it additionally samples a Bernoulli random variable $X_i$ with parameter $p = p(\lambda_m(t_i)) = \lambda_m(t_i)/\lambda_{\max}$. Finally, it accepts all the events $t_i$ such that $X_i = 1$. *i.e.*, $\mathcal{H}_m = \{t_i \in \mathcal{H}_{\max} \,|\, X_i = 1\}$. Here, note that the specific choice of $\lambda_{\max}$ does not affect the distribution of accepted events as long as $\lambda_{\max} \geq \lambda_m(t)$ for all $m \in \mathcal{M}$ and $t \in \mathbb{R}^+$.[7] Algorithm 4 in Appendix B.1 summarizes the overall procedure, where the parameter $p$ is often called the thinning probability.

Given a process of interest with intensity $\lambda_m(t)$, Lewis' thinning algorithm is helpful to make predictions about future events. For example, it can be used to compute Monte Carlo estimates of the average number of events $\mathbb{E}[N(t)]$ at a time $t$ in the future. However, it is not sufficient to make counterfactual predictions, *e.g.*, given the sequences of events $\mathcal{H}_{\max}$ and $\mathcal{H}_m$, we cannot know what would have happened if, at time $t_i$, the intensity had been $\lambda_{m'}(t_i)$, with $m \neq m'$, instead of $\lambda_m(t_i)$. To overcome this limitation, we will now augment the above thinning algorithm using a Gumbel-Max SCM. More specifically, let $\mathcal{C}$ be a SCM defined by the assignments:

$$X_i = \operatorname*{argmax}_{x \in \{0,1\}} g(x, \Lambda_i, \boldsymbol{U}_i) \quad \text{and} \quad \Lambda_i = \lambda(t_i), \tag{4}$$

where

$$g(x, \Lambda_i, \boldsymbol{U}_i) = \log p(X_i = x \,|\, \Lambda_i) + U_{i,x},$$

with $p(X_i = x \,|\, \Lambda_i) = x\, p(\Lambda_i) + (1-x)\,(1 - p(\Lambda_i))$, $p(\Lambda_i) = \Lambda_i/\lambda_{\max}$, $U_{i,x} \sim \text{Gumbel}(0, 1)$, and $t_i \sim \lambda_{max}$. Then, the thinning probabilities in Lewis' thinning algorithm are given by the following interventional distributions over $\mathcal{C}$[8]:

$$P^{\mathcal{C}\,;\,\mathrm{do}(\Lambda_i = \lambda(t_i))}(X_i = 1) = p(\lambda(t_i)) = \frac{\lambda(t_i)}{\lambda_{\max}}. \tag{5}$$

Under this view, given a sequence of accepted events $\mathcal{H}_m$ and rejected events $\mathcal{H}_{\max} \backslash \mathcal{H}_m$ under an intensity $\lambda_m(t)$, as determined by the binary samples $\{x_i\}$, we can estimate the posterior distribution $P^{\mathcal{C}\,|\, X_i = x_i, \Lambda_i = \lambda_m(t_i)\,;\,\mathrm{do}(\Lambda_i = \lambda_{m'}(t_i))}(U_{i,x})$ of each Gumbel noise variable $U_{i,x}$ using an efficient procedure [48] (refer to Appendix B.3). Then, we can use these noise posteriors to compute an unbiased finite sample Monte-Carlo estimate of the counterfactual thinning probability, *i.e.*,

$$P^{\mathcal{C}|X_i = x_i, \Lambda_i = \lambda_m(t_i)\,;\,\mathrm{do}(\Lambda_i = \lambda_{m'}(t_i))}(X_i = x) = \mathbb{E}_{\boldsymbol{U}_i | X_i = x_i, \Lambda_i = \lambda_m(t_i)}[\mathbf{1}[x = \operatorname*{argmax}_{x' \in \{0,1\}} g(x', \lambda_{m'}(t_i), \boldsymbol{U}_i)]],$$

where we drop $\mathrm{do}(\cdot)$ because $\boldsymbol{U}_i$ and $\lambda_m(t_i)$ are independent in the counterfactual SCM. Importantly, the above counterfactual thinning probability allows us to make counterfactual predictions, *e.g.*, given

---

[7]In this context, note that, rather than using an homogeneous Poisson process with intensity $\lambda_{\max}$, one could use any process with (time-varying) intensity $\lambda'(t) \geq \lambda_m(t)$ for all $m \in \mathcal{M}$ and $t \in \mathbb{R}^+$, as shown in Theorem 1 in Lewis and Shedler [20].

[8]This equality follows immediately from the Gumbel-Max *trick* [47]

**Algorithm 2** It samples a counterfactual sequence of events given a sequence of observed events from an inhomogeneous Poisson process.

---
1: **Input**: $\lambda_m(t), \lambda_{m'}(t), \mathcal{H}_m, \lambda_{\max}, T$.
2: **Initialize**: $\mathcal{H}_{m'} = \emptyset$.

3: **function** CF$(\lambda_m(t), \lambda_{m'}(t), \mathcal{H}_m, \lambda_{\max}, T)$
4: $\quad \mathcal{H}_{\max}, \_ \leftarrow$ LEWIS$(\lambda_{\max} - \lambda_m(t), \lambda_{\max}, T)$
5: $\quad \mathcal{H}_{\max} \leftarrow \mathcal{H}_{\max} \cup \mathcal{H}_m$
6: $\quad \mathcal{H}_{m'} \leftarrow$ ACC$(\lambda_m(t), \lambda_{m'}(t), \mathcal{H}_m, \mathcal{H}_{\max}, \lambda_{\max})$
7: $\quad$ **return** $\mathcal{H}_{m'}$
8: **end function**

---

a sequence of accepted events $\mathcal{H}_m$ and rejected events $\mathcal{H} \backslash \mathcal{H}_m$ under intensity $\lambda_m(t)$, we can use the counterfactual thinning probability to predict which events, among those in $\mathcal{H}$, would have been accepted if the intensity had been $\lambda_{m'}(t)$ instead of $\lambda_m(t)$. Algorithm 1 summarizes the algorithm. Here, it is important to note that the specific choice of $\lambda_{\max}$ does not affect the distribution of counterfactual events as long as $\lambda_{\max} \geq \lambda_m(t)$ for all $m \in \mathcal{M}$ and $t \in \mathbb{R}^+$, similarly as in standard thinning (refer to Appendix A.1 for a proof).

Finally, it is important to note that, by using Gumbel-Max SCMs, our causal model of thinning satisfies the monotonicity assumption [46, 21], discussed in Section 2, and thus the counterfactual thinning probability does not suffer from non-identifiability issues. More formally, we have the following proposition (proven in Appendix A.2):

**Proposition 1** *Let $\mathcal{M} = \{\lambda_m(t), \lambda_{m'}(t)\}$ and $\mathcal{C}$ be the corresponding causal model of thinning. Then, if $\lambda_m(t_i) \geq \lambda_{m'}(t_i)$, it holds that $P^{\mathcal{C} \,|\, X_i=0, \Lambda_i=\lambda_m(t_i)\,;\,do(\Lambda_i=\lambda_{m'}(t_i))}(X_i = 1) = 0$. Conversely, if $\lambda_m(t_i) \leq \lambda_{m'}(t_i)$, it holds that $P^{\mathcal{C} \,|\, X_i=1, \Lambda_i=\lambda_m(t_i)\,;\,do(\Lambda_i=\lambda_{m'}(t_i))}(X_i = 0) = 0$.*

The above result directly implies that, if a potential event $t_i \in \mathcal{H}_{\max}$ was rejected under $\lambda_m(t)$, *i.e.*, $t_i \notin \mathcal{H}_m$, then in a counterfactual scenario, if $\lambda_{m'}(t) \leq \lambda_m(t)$, it is *impossible* that the event is accepted under $\lambda_{m'}(t)$, *i.e.*, $t_i \in \mathcal{H}_{m'}$. Conversely, if a potential event $t_i \in \mathcal{H}_{\max}$ was accepted under $\lambda_m(t)$, *i.e.*, $t_i \in \mathcal{H}_m$, then in a counterfactual scenario, if $\lambda_{m'}(t) \geq \lambda_m(t)$, it is *impossible* that the event is rejected under $\lambda_{m'}(t)$, *i.e.*, $t_i \notin \mathcal{H}_{m'}$.

## 4 Sampling Counterfactual Events

In this section, we develop a sampling algorithm that, given an observed realization from a temporal point process with a given intensity function, it uses Algorithm 1 and the superposition theorem [22] to generate counterfactual realizations of the temporal point process under a given alternative intensity function. To ease the exposition, we first focus on inhomogeneous Poisson processes and then generalize our algorithm to linear Hawkes processes.

**Inhomogenous Poisson processes.** Assume we have observed a sequence of real events $\mathcal{H}_m$ and this sequence can be accurately characterized, observationally, using an inhomogeneous Poisson process with intensity $\lambda_m(t) = g(t)$, where $g(t) \geq 0$ is a time-varying function. In reality, this sequence of events has not been *generated* by Lewis' thinning algorithm but by a natural phenomena of interest. As a result, we cannot directly apply Algorithm 1 since it requires both a sequence of accepted and rejected events. However, we can find plausible sequences of events that Lewis' thinning algorithm would have rejected if it had accepted the observed sequence of events.

By construction, the intensity of accepted and rejected events $\lambda_{\text{accepted}}(t)$ and $\lambda_{\text{rejected}}(t)$ provided by Lewis' thinning algorithm should satisfy that $\lambda_{\max} = \lambda_{\text{accepted}}(t) + \lambda_{\text{rejected}}(t)$. Then, if $\lambda_{\text{accepted}}(t) = \lambda_m(t)$, $\lambda_{\max} \geq \max_t \lambda_m(t)$ and $\mathcal{H}_{\text{accepted}} = \mathcal{H}_m$, by the superposition theorem, we can find plausible sequences of events that Lewis' algorithm would have rejected just by sampling from the intensity $\lambda_{\text{rejected}}(t) = \lambda_{\max} - \lambda_m(t)$. Then, these generated sequences of rejected events, together with the sequence of observed events, can be fed into Algorithm 1 to sample sequences of counterfactual events given an alternative intensity $\lambda_{m'}(t)$. Algorithm 2 summarizes the algorithm, where LEWIS$(\cdot)$ samples a sequence of events using Algorithm 4 in Appendix B.1 and ACC$(\cdot)$ samples a counterfactual sequence of accepted events using Algorithm 1.

**Linear Hawkes processes.** Assume we have observed a sequence of real events $\mathcal{H}_m$ and it can be accurately characterized, observationally, using a linear Hawkes process with an intensity $\lambda_m(t)$

**Algorithm 3** It samples a counterfactual sequence of events given a sequence of observed events from a Hawkes process.

---
1: **Input**: $\mu_m, \alpha_m, g_m(t), \mu_{m'}, \alpha_{m'}, g_{m'}(t), \mathcal{H}_m, \lambda_{\max}, T$.
2: **Initialize**: $\mathcal{H}_{m'} = \emptyset$.

3: $\{\mathcal{H}_{m,j}\} \leftarrow \text{ASSIGN}(\mathcal{H}_m, \lambda_m(t))$
4: $\mathcal{H}_{m',0} \leftarrow \text{CF}(\gamma_{m,0}(t), \gamma_{m',0}(t), \mathcal{H}_{m,0}, \lambda_{\max}, T)$
5: $\mathcal{H}_{m'} \leftarrow \mathcal{H}_{m'} \cup \mathcal{H}_{m',0}$

6: **for** $t_j \in \mathcal{H}_m$ **do**
7:   **if** $t_j \in \mathcal{H}_{m'}$ **then**
8:     $\mathcal{H}_{m',j} \leftarrow \text{CF}(\gamma_{m,j}(t), \gamma_{m',j}(t), \mathcal{H}_{m,j}, \lambda_{\max}, T)$
9:     $\mathcal{H}_{m'} \leftarrow \mathcal{H}_{m'} \cup \mathcal{H}_{m',j}$
10:   **end if**
11: **end for**

12: $\mathcal{H} \leftarrow \mathcal{H}_{m'} \backslash \mathcal{H}_m$
13: **while** $|\mathcal{H}| > 0$ **do**
14:   $t_k \leftarrow \min_{t \in \mathcal{H}} t$
15:   $\mathcal{H}_{m,k}, \_ \leftarrow \text{LEWIS}(\gamma_{m',k}(t), \lambda_{\max}, T)$
16:   $\mathcal{H} \leftarrow \mathcal{H}_{m,k} \cup \mathcal{H} \backslash \{t_k\}$
17:   $\mathcal{H}_{m'} \leftarrow \mathcal{H}_{m'} \cup \mathcal{H}_{m,k}$
18: **end while**
19: **return** $\mathcal{H}_{m'}$

---

parameterized by $\mu_m$, $\alpha_m$ and $g_m(\cdot)$, as defined in Eq. 2. If our goal is to sample sequences of counterfactual events $\mathcal{H}_{m'}$ given an alternative Hawkes intensity $\lambda_{m'}(t)$ parameterized by $\mu_{m'}$, $\alpha_{m'}$ and $g_{m'}(\cdot)$, we cannot proceed similarly as in the case of inhomogeneous Poisson processes. This is because, to sample plausible rejected events within Algorithm 2, we need to pick a value of $\lambda_{\max}$ that upper bounds both $\lambda_m(t)$ and $\lambda_{m'}(t)$, otherwise, Algorithm 1 might break because the thinning probabilities $p$ used by the causal model of thinning could be greater than 1. Unfortunately, since $\lambda_{m'}(t)$ depends on the counterfactual history $\mathcal{H}_{m'}$ we aim to sample, we cannot know its maximum value at the time we sample the rejected events. However, we can overcome this challenge by resorting to the branching process interpretation of Hawkes processes [24].

More specifically, we can view any linear Hawkes process as a superposition of several temporal point processes, *i.e.*, a process with constant intensity $\gamma_0(t) = \mu$ and, for each event $t_i \in \mathcal{H}$, a process with intensity $\gamma_i(t) = \alpha g(t - t_i)$. Under this view, we can naturally derive the following thinning algorithm to sample from linear Hawkes processes [49]. First, we sample a sequence of events $t_i$ from the process with intensity $\gamma_0(t)$. Then, for each sampled event $t_i$, we create a process with intensity $\gamma_i(t)$ and sample a sequence of events $t_j$ independently from each of them using Algorithm 4 in Appendix B.1. Further, for each of these sampled events $t_j$, we again create another set of processes with intensity $\gamma_j(t)$ and sample from them independently, and continue recursively. By the superposition theorem, it readily follows that the overall sequence of sampled events is a valid realization of the original Hawkes process. Algorithm 5 in Appendix B.2 summarizes the algorithm.

Importantly, all the intensities $\gamma_j(t)$ we sample from are bounded by $\max\{\mu, \max_t \alpha g(t)\}$. As a result, given two intensities of interest $\lambda_m(t)$ and $\lambda_{m'}(t)$, we can sample sequences of counterfactual events by running Algorithm 2 independently for each of the above processes with

$$\lambda_{\max} \geq \max\{\mu_m, \mu_{m'}, \max_t \alpha_m g_m(t), \max_t \alpha_{m'} g_{m'}(t)\}.$$

However, to do so, we also need to assign each observed event $t_i \in \mathcal{H}_m$ to one of the above processes with probability $\gamma_{m,j}(t_i) / \sum_{k<i} \gamma_{m,k}(t_i)$, which is the probability that the process has *caused* the event [50]. Algorithm 3 summarizes the resulting algorithm, where $\text{ASSIGN}(\cdot)$ returns the observed events $\mathcal{H}_{m,j}$ assigned to each process $j$, $\text{CF}(\cdot)$ samples a counterfactual sequence of events using Algorithm 2, and $\text{LEWIS}(\cdot)$ samples a sequence of events using Algorithm 4 in Appendix B.1. Within the algorithm, it is also worth noting that lines 6-11 need to go through the observed events $t_j$ in chronological order and, for each $t_j$, the algorithm only accepts counterfactual events from the corresponding process with intensity $\gamma_{m',j}(t)$ if the event $t_j$ has been previously accepted in the counterfactual realization. Moreover, lines 12-18 sample a sequence of events for each of the processes triggered by the counterfactual events that did not exist in the observed sequence of events.

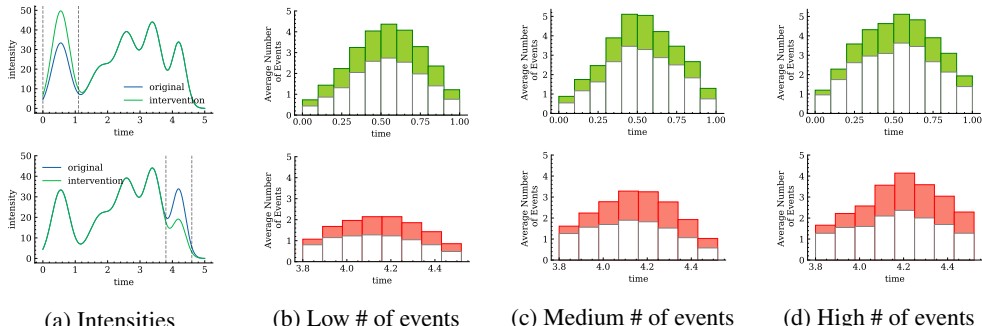

(a) Intensities     (b) Low # of events     (c) Medium # of events     (d) High # of events

Figure 1: Effect of interventions in two inhomogeneous Poisson processes. Panel (a) shows the intensities of the original and the intervened processes (in blue and green, respectively) and a window of interest (dashed vertical lines). Panels (b-d) show the difference in average number of events over time between the counterfactual and the original realizations, where each row corresponds to one process and we group the original realizations in three quantiles according to their overall number of events in the time window of interest. Positive (negative) differences are shown in green (red). In each experiment, we sample 1,000 realizations from the original process and, for each of these realizations, we sample 100 counterfactual realizations from the intervened process.

## 5 Experiments on Synthetic Data

In this section, we feed Algorithms 2 and 3 with realizations of synthetic inhomogeneous Poisson processes and linear Hawkes processes and investigate to what extent the counterfactual realizations under alternative intensity functions differ from the original realizations fed to them[9].

**Experimental setup.** We consider the family of inhomogeneous Poisson processes $\mathcal{M}(\boldsymbol{\phi}, \boldsymbol{\alpha}, \boldsymbol{\tau})$ parameterized by weighted combinations of RBF kernels, *i.e.*, $\lambda(t) = \sum_j \phi_j \exp\left(-\alpha_j(t - \tau_j)\right), \ t \geq 0$, where $\phi_j, \alpha_j, \tau_j \geq 0$, and the family of linear Hawkes processes $\mathcal{M}(\mu, \alpha, \omega)$ defined in Eq. 2, with exponential triggering kernels $g(t) = \exp(-\omega t)$. Moreover, we experiment with simple interventions under which, for inhomogeneous Poisson processes, one of the RBF kernels, picked at random, change its amplitude, *i.e.*, $\phi_{m',i} = \max(\phi_{m,i} + \epsilon, 0)$, and, for Hawkes processes, the parameter $\alpha$ change its value, *i.e.*, $\alpha_{m'} = \max(\alpha_m + \epsilon, 0)$, where $\epsilon \sim N(0, \sigma)$.

In each experiment, we first sample 1,000 realizations from a process with one set of parameters using Algorithm 4 (or Algorithm 5). Then, we carry out the above mentioned intervention and, for each of the sampled realizations, we use Algorithm 2 (or Algorithm 3) to sample 100 counterfactual realizations under the resulting alternative set of parameters. Finally, we partition the original realizations in three quantiles according to their overall number of events in a time window of interest and, for each quantile, we look at the number of events in the corresponding counterfactual realizations in the same window of interest. In all experiments, Algorithms 1—3 use 100 samples from the posterior distribution $P^{\mathcal{C} \,|\, X_i = x, \Lambda_i = \lambda(t_i) \,;\, \mathrm{do}(\Lambda_i = \lambda_{m'}(t_i))}(U_i)$ of each Gumbel noise variable $U_{i,x}$ to estimate the counterfactual thinning probabilities $P^{\mathcal{C} \,|\, X_i = x, \Lambda_i = \lambda(t_i) \,;\, \mathrm{do}(\Lambda_i = \lambda_{m'}(t_i))}(X_i)$.

**Results.** Figure 1 summarizes the results for two specific inhomogeneous Poisson processes undergoing one of the above mentioned interventions—we found qualitatively similar results for other inhomogeneous Poisson processes. In the top row, the results show that, under our model, an intervention that increases the original intensity of the process, by increasing the amplitude of one of the RFB kernels by approximately an additional half does not have the same effect across realizations. In realizations with a low (high) number of events, the average number of counterfactual events in the window of interest increases up to $\sim 60\%$ ($\sim 40\%$) over time with respect to the average number of events in the original realizations. In the bottom row, we find that this is also true for an intervention that decreases the original intensity of the process, by approximately halving the amplitude of another of the RBF kernels. However, the difference is smaller in relative terms.

Figures 2 summarizes the results for a specific Hawkes process undergoing two of the above mentioned interventions—we found qualitative similar results for other Hawkes processes. We find that,

---

[9]All experiments were performed on a machine with 48 Intel(R) Xeon(R) 3.00GHz CPU cores and 1.5TB.

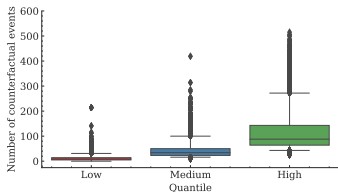

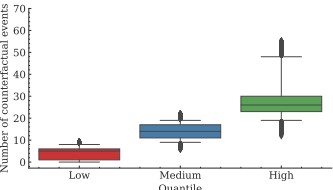

(a) Self-excitation increases

(b) Self-excitation decreases

Figure 2: Effect of interventions in Hawkes processes. Panels summarize the distribution of the number of events per counterfactual realization corresponding to original realizations with a low (red; $|\mathcal{H}_m| \in [0,9]$, $\mathbb{E}[|\mathcal{H}_m|] = 5.04$), medium (blue; $|\mathcal{H}_m| \in [10, 22]$, $\mathbb{E}[|\mathcal{H}_m|] = 17.17$) and high (green; $|\mathcal{H}_m| \in [25, 59]$, $\mathbb{E}[|\mathcal{H}_m|] = 36.17$) number of events. The horizontal lines within the boxes indicate average value, the boxes indicate 25% and 75% quantiles, the whiskers indicate 5% and 95% quantiles and the points are outliers. Here, we sample 1,000 realizations from the original process, with parameters $\mu_m = 1$, $\alpha_m = 1$, and $\omega_m = 1$, and, for each of these realizations, we sample 100 counterfactual realizations from each of the intervened process. In panel (a), the intervened process has parameter $\alpha_{m'} = 1.44$, in panel (b), it has parameter $\alpha_{m'} = 0.75$ and, in both panels, the remaining parameters $\mu_{m'} = \mu_m$ and $\alpha_{m'} = \alpha_m$. We set the time horizon $T = 5$.

similarly as in inhomogeneous Poisson processes, the interventions do not have the same effect across realizations. However, in this case, the difference among them is more stark—while the average number of counterfactual events (a) increases by 115% and (b) decreases by 11% for realizations with a low number of events, it (a) increases by 216% and (b) decreases by 21% for realizations with a high number of events. Moreover, we also find that, there is a high variability across counterfactual realizations, especially when $\alpha_{m'} > \alpha_m$. For example, while the original realizations with a low number of events never contained more than 9 events, there exists counterfactual realizations with more than 200 events. This is due to the self-exciting property of Hawkes processes by which counterfactual events may trigger the emergence of additional counterfactual events, shown as yellow dots in Figure 7 in Appendix C.

## 6 Experiments on Epidemiological Data

In this section, we run a (simple) variation of Algorithm 3 (refer to Appendix B.4) to quantify the effect of interventions on a networked Susceptible-Infectious-Recovered (SIR) epidemiological model [51] fitted using real event data from an Ebola outbreak in West Africa in 2013-2016 [52].

**Experimental setup.** We build upon the networked Susceptible-Infectious-Recovered (SIR) epidemiological model introduced by Lorch et al. [51], which is based on temporal point processes. Given a contact network $\mathcal{G} = (\mathcal{V}, \mathcal{E})$, we represent the times when each node gets infected and recovered using a collection of binary counting processes $\boldsymbol{Y}(t)$ and $\boldsymbol{W}(t)$ and we track the current state of each node using a collection of state variables $\boldsymbol{X}(t) = \boldsymbol{Y}(t) - \boldsymbol{W}(t)$, where $X_i(t) = 1$ indicates node $i \in \mathcal{V}$ is infected at time $t$ and $X_i(t) = 0$ indicates it is susceptible or recovered. For each node $i \in \mathcal{V}$, we characterize the above counting processes using the following intensities

$$\mathbb{E}[dY_i(t)\,|\,\mathcal{H}(t)] = (1 - X_i(t)) \sum_{j\,|\,(i,j)\in\mathcal{E}} \beta\, X_j(t)dt \quad \text{and} \quad \mathbb{E}[dW_i(t)\,|\,\mathcal{H}(t)] = \delta\, X_i(t)dt, \quad (6)$$

where note that the counting process $\boldsymbol{Y}(t)$ can be viewed as a (networked) multidimensional Hawkes process with stochastic triggering kernels defined by step functions. Refer to Appendix D for more details on how we set the parameters $\beta$ and $\delta$ and how we generate the contact network $\mathcal{G}$.

In our experiments, we simulate a *realistic* outbreak by sampling a realization from the above fitted model. To this end, for each real recorded case up to January 1, 2014 [52], we sample a seed node at random from the same district as the observed case[10]. Figure 9 in Appendix D shows the geographical distribution of seed infections and overall cumulative number of infections in the outbreak. Then, given this sampled realization, we quantify the effect of two types of interventions by sampling counterfactual realizations using a variation of Algorithm 3 (refer to Appendix B.4):

---

[10]The first real case was recorded on Dec 26, 2013. By January 1, 2014, there were six recorded cases in four different districts.

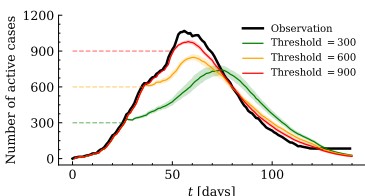

(a) Contacts reduction in highest incidence district

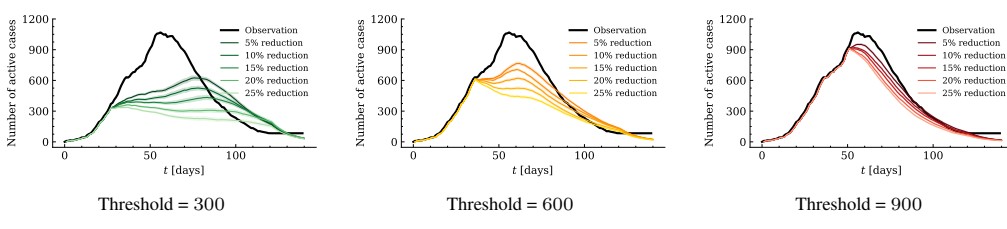

(b) Contacts reduction in all districts

Figure 3: Effect of interventions where individuals reduce their contacts after the overall number of active infections reaches a varying threshold. In all figures, the black line ("Observation") corresponds to an outbreak sampled from the fitted SIR model, the remaining lines correspond to counterfactual realizations for this outbreak under different interventions, and the shaded regions correspond to 95% confidence intervals. For each threshold and reduction level, we repeat the experiment five times.

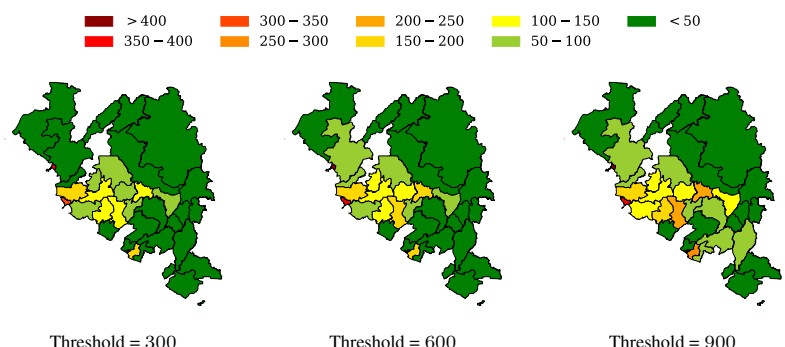

Figure 4: Geographical distribution of the overall cumulative number of infections per district in counterfactual outbreaks where all individuals reduce their individual contacts by 25% after the overall number of active infections reaches a certain threshold.

— *Reduction of number of contacts*: individuals reduce their individual contacts after the overall number of active infections reaches a certain threshold. In one scenario, only individuals from the district with the highest incidence reduce their contacts within the district and get isolated from all other districts and, in another scenario, everyone reduces their individual contacts.
— *Vaccination*: a percentage of the overall population receives a vaccine with a certain level of efficacy. Within our model, we measure vaccine efficacy in terms of reduction of the value of the parameter $\beta$, which controls the infection rate between individuals.

In each experiment, to estimate the average and confidence intervals of the outcome of interest (*e.g.*, number of cases), we sample 20 counterfactual realizations.

**Results.** Figures 3–4 summarize the results for the interventions where individuals reduce their individual contacts after the overall number of active infections reaches a certain threshold. Our results suggest that, at all threshold levels, reducing the individual contacts across all districts, even by just 5%, would have been more effective than isolating and reducing the contacts by 50% in the district with higher incidence. Moreover, we also find that, for lower threshold values, the counterfactual outbreaks would have spread to fewer districts and the overall number of infections would have been significantly lower. Figures 5–6 summarize the results for the interventions where a percentage of the overall population receives a vaccine with a certain level of efficacy. Our results suggest that a high level of vaccine effectivity would have not been sufficient to reduce the number of infections if

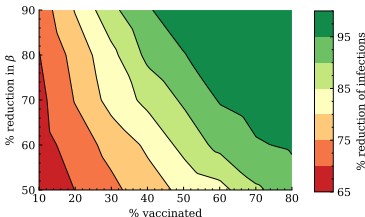

Figure 5: Effect of interventions where a percentage of the overall population receives a vaccine with a certain level of efficacy. The figure shows the average reduction in the cumulative number of infections under each intervention with respect to an outbreak sampled from the fitted SIR model.

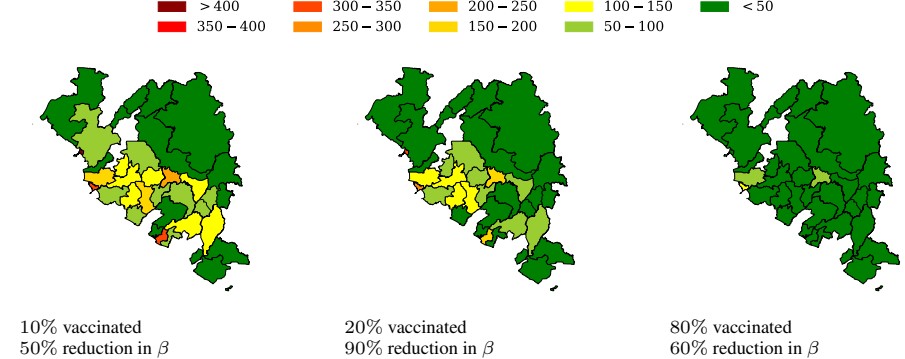

10% vaccinated
50% reduction in $\beta$

20% vaccinated
90% reduction in $\beta$

80% vaccinated
60% reduction in $\beta$

Figure 6: Geographical distribution of the overall cumulative number of infections per district tin counterfactual outbreaks where a percentage of the population receives a vaccine with a certain level of efficacy.

the percentage of the population who had received the vaccine was *low*. For example, if less than 20% of the population had received the vaccine, even a vaccine with a 90% effectivity would have been unable to reduce the infections by >70%. If >80% of the population had received the vaccine, a vaccine with just a 60% effectivity would have reduced the infections by >95%. Finally, we find that, similarly as in the scenario where individuals reduce their individual contacts, the reductions in the overall number of infections would have also led to lower geographical dispersion.

## 7 Conclusions, Limitations and Future Work

Since counterfactual reasoning lies within level three in the "ladder of causation" [45], we have been unable to validate our counterfactual predictions using observational nor interventional experiments. That being said, we have made an intuitive assumption—monotonicity—about the causal mechanism of the world—our causal model of thinning—which specifies how changes on the intensity function of a temporal point process may have lead to particular outcomes while holding "every-thing else" fixed, similarly as previous work [21, 41]. In this context, it would be worth to understand the sensitivity of counterfactual realizations to the specific choice of SCM. In our work, we have focused on temporal point processes, however, some of our ideas can be readily extended to spatial point processes. Moreover, it would be interesting to extend our methodology to support nonlinear Hawkes processes and neural Hawkes processes [53, 54]. To this end, the main challenge is that there is no branching process interpretation for nonlinear or neural Hawkes and thus we would have to find alternative ways to bound the value of the relevant intensity functions. Finally, it would be important to carry out a user study in which the counterfactual realizations provided by our algorithm are shared with domain experts (*e.g.*, epidemiologists) and evaluate our sampling algorithm using other real datasets from other applications such as climate change, recommendation systems and voting data.

## Acknowledgments and Disclosure of Funding

We would like to thank William Trouleau for sharing with us a pre-processed version of the Ebola dataset as well as the fitted stochastic block model we used in our simulation experiments and thank Stratis Tsirtsis and Abir De for fruitful discussions. Gomez-Rodriguez acknowledges support from the European Research Council (ERC) under the European Union's Horizon 2020 research and innovation programme (grant agreement No. 945719).

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
