# OpenReview forum: "Counterfactual Temporal Point Processes"
_NeurIPS.cc/2022/Conference — NeurIPS 2022 Accept_

### Official Review · Reviewer_Ddj9 · 2022-07-05

**Rating:** 6
**Confidence:** 3
**Soundness:** 2 fair
**Presentation:** 2 fair
**Contribution:** 3 good

**Summary:**

This paper proposes a causal model of thinning for temporal point processes. Given a set of observed events for a point process one can use the superposition theorem to sample plausible sequences of rejected events. Given the observed events and the rejected events, a SCM can be used to sample counterfactual events. These are realisations of the point process we would observe under alternative intensity functions. A Gumbel-Max SCM is used to ensure identifiability of the counterfactual distributions.

**Questions:**

See section above.

**Limitations:**

Yes, I cannot identify potential negative societal impact for this work.


**Strengths And Weaknesses:**

The paper is well written and easy to follow. This is one of the few papers I have read combining point processes with causality which makes it novel and interesting despite being quite simple. The paper would benefit from clarifying the points given below:

- Section 3. I found the last part of this section (from line 153), which I believe to be the most important from a causality point of view, quite confusing. Given that the authors use the language of structural causal models to frame the problem, it would be useful to add a causal graph representing the data generating mechanism. This is important as it determines how the equation under line 157 is computed and what type of adjustment the do-calculus requires. For instance do(.) is dropped ‘because U and $\lambda$ are independent in the counterfactual SCM'. This is not evident from your discussion and it could be clarified using causal graphs. In addition, a short sudo algorithm describing how to sample from the distribution after line 157 would clarify the paper.
- Line 155. How you estimate p(U) is very important in this paper as it also justifies the choice of the Gumble max SCM. This should be explained in the preliminaries or in the appendix.
- Section 4. Depending on the value of $\lambda_{max}$ the number of rejected events consistent with $\lambda_m$ changes. How would that change the counterfactual realization that you would generate from Alg 1? Would that change the distribution of $U_i$ and therefore the distribution at line 7 of the algorithm?
- In Alg 1 the function ACC(.) takes $\lambda_{max}$ as input, where is that used? I don't think you need it once you have $H_{max}$
- Could you clarify why we need to restrict the SCM to be the Gumbel max. Is this to ensure that given the data from observational distribution we are sure that we can identify the counterfactual distribution correctly?
- What is preventing you from working with points in a higher dimensional space? What about spatial point processes for example?

*Experimental section:*
- I liked the application the authors look at in this paper as it highlights the importance of the problem addressed. However, assessing the behavior of the proposed methodology with real-world data is difficult. I would prefer to see at least one experiment for the synthetic settings in the main paper.
- Given that you assume the intensity to be known, the only approximations you are introducing are the estimation of the distribution of the noise variables and the Monte-Carlo approximation of the counterfactual thinning probability. Given these approximations, I wonder if one could assess how close the generated counterfactual realizations are to the ground truth. In a synthetic setting you can obtain counterfactual realizations from a known SCM and compare them with those you generate.

*Minor:*
Line 150 what do you mean by $t_i \sim \lambda_{max}$? I think in thinning the total number of points are sampled from a Poisson and then the points are uniformly distributed?

---

> ### Author Response · Authors · 2022-07-27
> **Response to Reviewer Ddj9**
>
> We thank the reviewer for the thoughtful review and constructive feedback. We have updated the main paper accordingly and, in what follows, we provide a point-by-point response. If the paper gets accepted, we will also update the Appendix with an explanatory causal graph, an additional algorithm box and a sensitivity analysis as promised below.
>
> **[Adding the casual graph representing the mechanism]**
>
> We will add a causal graph representing the data generating mechanism in the revised version of the Appendix.
>
> **[Estimating / sampling from the noise posterior]**
>
> We will include an algorithm box summarizing the efficient procedure [19, 43] our method relies on to estimate / sample from the noise posterior in the revised version of the Appendix.
>
> **[Effect of $\lambda_\text{max}$ in counterfactual realizations]**
>
> It is true that the number of rejected events to consider as well as the noise posterior vary with $\lambda_{\text{max}}$. However, the actual distribution of counterfactual events generated by Algorithm 1 is **invariant** to the choice of $\lambda_{\text{max}}$ as discussed in lines 162-165 (168-170) of the original (revised) submission and formally proven in Appendix A.
>
> **[Use of $\lambda_\text{max}$ in Algorithm 1]**
>
> The function ACC() in Algorithm 1 uses $\lambda_\text{max}$ to compute the counterfactual thinning probability in line 7. More specifically, the counterfactual thinning probability depends on the posterior of the noise, which depends on $g(...)$, which depends on $\lambda_\text{max}$.
>
> **[Why Gumbel Max SCM?]**
>
> We use the Gumbel-Max because, in our setting, it satisfies the monotonicity assumption (Proposition 1) and, as a consequence, the counterfactual intensity does not suffer from non-identifiability issues, as discussed in lines 116-125 (121-124) and 166-169 (171-174) of the original (revised) submission.
>
> **[Spatial point processes]**
>
> Our method could be readily extended to spatial inhomogeneous Poisson point processes and spatial linear Hawkes processes. However, for ease of exposition, we focus on temporal point processes. We have clarified this in line 310-311 of the revised submission.
>
> **[Add synthetic experiments to the main paper]**
>
> If our submission is accepted, we will move (part of the) synthetic experiments from Appendix D to the main using the additional content page allowed for the camera-ready version. We could not move it in the revised version of the paper submitted during the rebuttal period due to lack of space.
>
> **[Compare how close counterfactuals are to the ground truth]**
>
> We would first like to note that, under our model, there is not an analytical expression for the counterfactual thinning probability. As a result, we cannot compare the generated counterfactual realizations to a ground truth. That being said, for completeness, we will add a sensitivity analysis of the Monte-Carlo approximation of the counterfactual thinning probability in the revised version of the Appendix.
>
> **[Line 150, what we mean by $t_i \sim \lambda_\text{max}$]**
>
> In line 150 (155) of the original (revised) submission, $t_i \sim \lambda_\text{max}$ means that the event $t_i$ is sampled from an inhomogeneous Poisson process with intensity $\lambda_\text{max}$. It is true that one can sample from an inhomogeneous Poisson process in the way the reviewer describes. However, in our implementation, we sample each event in chronological order by sampling the inter event time between consecutive events from an exponential distribution with rate $\lambda_\text{max}$.

---

### Official Review · Reviewer_E24B · 2022-07-12

**Rating:** 7
**Confidence:** 4
**Soundness:** 3 good
**Presentation:** 2 fair
**Contribution:** 3 good

**Summary:**

This paper proposes an approach to address counter-factual queries in certain types of temporal point processes, such as what would have happened if the conditional intensity was different in the past. This is a query that has not been well studied in TPPs, due to several technically challenging issues. The proposed work essentially relies on leveraging the thinning algorithm together with a Gumbel-Max structural causal model. There is also a sampling algorithm that uses this causal model of thinning together with the superposition theorem to simulate counterfactual realizations of TPPs. A limited application is studied in the main text for experiments.

**Questions:**

I have posed some questions previously but here are a few more questions and comments:

Could the authors please clarify that the proposed ideas work for univariate TPPs? What classes of TPPs are covered by the approach?

Why is [4] cited on line 21?

The authors make an incorrect distinction in the second paragraph of Section 1. How does it matter whether a machine makes a decision or a human? And how does it matter if the stakes are “high” or “low”? A decision is a decision, and I do not understand how counterfactuals are needed for “high-stake” decisions but not “low-stake” decisions. This entire paragraph needs to be rewritten, in my view. And the authors need to clearly explain the value of having a model that enables counterfactual queries.

There are several concepts introduced on page 2 that would not be easily understood by reader not familiar with TPPs.

I’m surprised that Pearl’s work and related work is not cited at all in Section 1. It seems to be left for much later.

g_i should probably be f_i on line 118.

Monotonicity is not clearly explained on page 3.

The Gumbel-Max trick mentioned in footnote 4 needs an explanation.

In lines 173-177, should “unlikely” be “impossible”, based on the probabilities in Proposition 1?

In my opinion, the application studied is unnecessarily involved, for what it ultimately illustrates. It is far from a real application and more like a case study that is mildly real-inspired. I understand of course that there is a general challenge around demonstrating counterfactual analysis, but I wonder if a variety of purely synthetic data experiments in the main text would have been more suitable.

**Limitations:**

The authors need to write a lot more clearly about the many limitations of the work.

**Strengths And Weaknesses:**

Strengths:

I think this paper has several new ideas for how to handle counterfactual queries in TPPs. It brings together previous ideas related to TPPs (such as thinning and superposition) and structural causal models (such as a recent Gumbel-Max version) but I think it does so in an innovative way. There are many non-trivial aspects in my view, such as how to handle the fact that “rejected” events from the thinning algorithm are not seen in the observational data. Thus I believe the work is sufficiently novel and significant.

Although there are a few details in some sections which I could not completely follow, I’m generally favorable towards the paper. I also however have some concerns and many suggestions, which I outline below.

Weaknesses (with questions and suggestions):

Firstly, the title of the paper is over-encompassing. There are several assumptions made, and as far as I understand, the approach does not necessarily work for all TPPs. A specific title that is more descriptive would be appropriate. I highly encourage the authors to propose and use a different title. For me, one of the major weaknesses of the work is the lack of detail around the limitations and assumptions (even beyond the title). The authors need to be clearer about the scope of the work and the implications of the assumptions.

A fundamental question for the authors: why do counterfactuals enable for better targeted interventions? Shouldn’t it be sufficient to have an interventional model if one is only interested in interventions? I found some of the motivation for the work on counterfactuals to be unfounded. Counterfactuals are clearly important for some sort of analysis, but I would like to better understand from the authors how it helps them in their problem setting.

I find some of the positioning of the related work to be unclear and even incorrect. For instance, Didelez (2008) is clearly related to Granger causality for multivariate TPPs, yet it is categorized in a separate bucket – I do not understand what “predicting quantities related to an interventional distribution” means. Is this a reference to average treatment effects? Could the authors please clarify? I recommend the authors refer to other work on Granger causal models as there is a lot of literature on graphical representations of TPPs. The ones that are cited are more neural network inspired than founded on graphical model foundations. Also, the authors should do a better job of mentioning why some of the prior work (such as by Roysland) is not directly relevant.

I found the experimentation section to be on the weaker side. I am surprised that synthetic experiments were placed in the Appendix, given that the paper is about counterfactual analysis. To me this is a sign that the scope of the work is quite narrow. This is not a major problem for me, but I think this should be made more explicit.

---

> ### Author Response · Authors · 2022-07-27
> **Response to Reviewer E24B**
>
> We thank the reviewer for the thoughtful review and constructive feedback. We have updated the paper accordingly and, in what follows, we provide a point-by-point response. Our response continues in the second comment.
>
> **[Title, limitations and assumptions]**
>
> In the revised submission, we have renamed Section 6 to “Conclusions, Limitations and Future Work''. This renamed section discusses the limitations of our method. In particular, it includes the remarks at the end of Section 5 in the original submission and a discussion on the challenges to extending our method to more complicated TPPs (please, refer to our answer to reviewer nA6T).
>
> If the reviewer still thinks that we should change the title even if the above mentioned limitations are summarized in Section 6, we would be happy to change it to, e.g., "Counterfactual Temporal Point Processes via a Causal Model of Thinning". We would be also open to any concrete suggestions the reviewer may have.
>
> **[Why counterfactual for better targeted interventions?]**
>
> We would like to first emphasize that our contribution is to develop a general method to generate counterfactuals in temporal point processes. As a potential motivating application, we use previous research on the cognitive science literature to argue that such counterfactuals may help humans implement better targeted interventions **given the outcome of a past intervention**. In particular, previous research [10, 11] has shown that counterfactuals help humans correct and improve behavior that **has been unsuccessful in the past**.
>
> To avoid misunderstandings, in the revised submission, we have rewritten: (i) the last sentence of the second paragraph in the introduction to: "In this work, our goal is to develop machine learning methods that, given the outcome of an intervention implemented in the past, are able to assist decision makers at implementing better interventions in these high-stakes applications."; and, (ii) the first sentence of the third paragraph in the introduction to “More specifically, we focus on facilitating counterfactual thinking, a type of thinking that has been argued to help humans correct and improve behavior that has been unsuccessful in the past”.
>
> **[Position of the related work]**
>
> We agree with the reviewer that Didelez (2008) has links to Granger causality and thus we have cited it together with [23-25] in the revised submission.
>
> Predicting quantities related to an interventional distribution refers to, e.g., average treatment effect (ATE), as the reviewer guessed, or conditional average treatment effect (CATE). We have clarified this in line 74 of the revised submission.
>
> We would be grateful if the reviewer can point us to specific pieces of work on Granger causal models founded on graphical model foundations and we will be happy to cite it in addition to [23-25].
>
> **[Synthetic experiments]**
>
> If our submission is accepted, we will move (part of the) synthetic experiments from Appendix D to the main using the additional content page allowed for the camera-ready version. We could not move it in the revised version of the paper submitted during the rebuttal period due to lack of space.
>
> **[Univariate TPPs / Classes of TPPs]**
>
> Our method allows for both univariate and multivariate inhomogeneous Poisson processes and linear Hawkes processes. In Appendix D, we perform experiments on synthetic data using **univariate** inhomogeneous Poisson processes and linear Hawkes processes. In Section 5, we perform experiments on real epidemiological data using multivariate linear Hawkes processes (as discussed in lines 249-250, the SIR model can be viewed as a **multivariate** linear Hawkes process).
>
> Please, refer to the answer to reviewer nA6T for a discussion on the challenges one would face to extend our method to more complicated TPPs such as neural Hawkes (e.g., Mei & Eisner, NeurIPS 2017) or nonlinear Hawkes.
>
> **[Why is [4] cited on line 21? ]**
>
> We cite Linderman and Adams [4] because they validate their proposed temporal point process model using stock market data.

---

> > ### Author Response · Authors · 2022-07-27
> > **Response to Reviewer E24B (Cont.)**
> >
> > **[Machine vs human decisions / High-stakes vs low-stakes]**
> >
> > We agree with the reviewer that one may apply our methodology to generate counterfactuals both in low- and high-stakes scenarios where decisions are taken by humans or machines. By highlighting a potential down-stream application of our methodology in the introduction (please, refer to our answer to ''[Why counterfactual for better targeted interventions]"), we did **not** mean to say that counterfactuals are not needed in low-stakes scenarios nor decisions taken by machines. Rather, we meant to provide one example of a potential application where there is empirical evidence that counterfactuals may be valuable [10, 11].
> >
> > In this context, it is also worth highlighting that, in the context of Markov decision processes, counterfactuals have been used to improve the training of reinforcement learning (RL) agents (e.g.,Buesing et al., ICLR 2019; Pitis et al., NeurIPS 2020). We have highlighted this in lines 86-87 of the revised submission.
> >
> > **[Concepts on page 2 related to TPPs]**
> >
> > Due to lack of space, on page 2, we give a brief summary of basic concepts in TPPs rather than a comprehensive tutorial. In this context, we refer the interested reader not familiar with TPPs to [39].
> >
> > **[In lines 173-177, should “unlikely” be “impossible”, based on the probabilities in Proposition 1?]**
> >
> > Yes, it should read impossible. We have changed this in the revised submission.

---

> > > ### Comment · Reviewer_E24B · 2022-08-08
> > > **Thanks for clarifications**
> > >
> > > I thank the authors for providing helpful clarifications and for being open to some suggestions (such as a proposed title change). I am open to increasing my score but I plan to wait until there are further discussions with other reviewers.

---

### Official Review · Reviewer_nA6T · 2022-07-12

**Rating:** 7
**Confidence:** 3
**Soundness:** 4 excellent
**Presentation:** 4 excellent
**Contribution:** 4 excellent

**Summary:**

In this paper, the authors proposed a simple but solid method to answer counterfactual questions for temporal point processes.
In particular, the authors focused on the counterfactual question that whether a historical event will happen if the corresponding intensity is changed.
For the inhomogeneous Poisson process, the authors demonstrated that via sampling events with a rejection-related intensity (i.e., lambda_max - lambda_accept) and superposing the events to observed ones, the counterfactual question can be answered simply based on a thinning process.
Moreover, leveraging the fact that Hawkes process can be implemented as the superposition of various inhomogeneous Poisson process, the proposed thinning process can be extended to answer the counterfactual question of linear Hawkes process — just applying the method to each branch of the Hawkes process.
A simulation experiment on epidemiological data demonstrates the rationality of the proposed method.

**Questions:**

No more questions.

**Limitations:**

(1) The method is focusing on inhomogeneous Poisson process and linear Hawks process. Whether it can be applied to other TPPs is unknown and needs to be discussed in depth.

(2) Besides epidemiological data, it would be nice if the authors can discuss more potential applications, e.g., recommendation systems, and voting data.

**Strengths And Weaknesses:**

Strength:
(1) The paper is well-written and the idea is easy to follow.
(2) The idea is simple and theoretically supported, and the method is easy to implement. Additionally, Proposition 1 fits our intuition.
(3) The application of epidemiological data is important and interesting.

Weaknesses:
(1) If my understanding is correct, the current application scenarios are limited to inhomogenous Poisson process and linear Hawkes process. It would be nice if the authors can discuss how to extend the proposed method to more complicated TPPs, e.g., neural Hawkes, or nonlinear Hawkes.

---

> ### Author Response · Authors · 2022-07-27
> **Response to Reviewer nA6T**
>
> We thank the reviewer for the thoughtful review and constructive feedback. We have updated the paper accordingly and, in what follows, we provide a point-by-point response.
>
> **[Extending our method to more complicated TPPs]**
>
> To extend our method to neural Hawkes or nonlinear Hawkes, one would need to bound the value of the alternative intensity function $\lambda'(t)$ under the intervention of interest to pick a feasible $\lambda_{\text{max}}$. In this context, note that Lewis' thinning algorithm can still be used to sample events from these more complicated TPPs (e.g., refer to Appendix B.3 in Mei & Eisner, NeurIPS 2017).
>
> For linear Hawkes processes, we could bound $\lambda'(t)$ using the branching process interpretation of linear Hawkes processes (refer to lines 210-234). However, such interpretation does not work for neural Hawkes and nonlinear Hawkes. Naively, one could just think of setting the $\lambda_{\text{max}}$ to a very high value, however, this would have an impact on the scalability of our method because the thinning process would reject many events. Therefore, more research is needed to efficiently extend our method to more complicated TPPs. In the revised submission, we have included a discussion about this in lines 310-314.
>
> **[Further practical applications (recommender systems and …)]**
>
> We have discussed more potential applications of our methodology in lines 317-318 of the revised submission, including recommendation systems and voting data.

---

### Official Review · Reviewer_7bXC · 2022-07-18

**Rating:** 4
**Confidence:** 4
**Soundness:** 2 fair
**Presentation:** 2 fair
**Contribution:** 2 fair

**Summary:**

The authors propose an approach for estimating counterfactuals in temporal point processes.

In the proposed approach, a temporal point process is modeled using the Gumbel-Max structural causal model under the assumptions of monotonicity and binary random variables. The error terms in this model are all independent of each other, which implies Markovianity, i.e., no unmeasured confounders). Under those assumption, counterfactuals are always identifiable.

The authors' contribution is a sampling algorithm based on the Lewis' thinning algorithm and the  superposition theorem. Lewis’ thinning algorithm is a popular method to sample from the a temporal point process. They authors use the superposition theorem to sample plausible sequences of events that the thinning process would have rejected if it had accepted the observed sequence of real events. The proposed sampling algorithm was evaluated using both synthetic and real epidemiological data.


**Questions:**

No questions.

**Limitations:**

In line 103, SCMs are defined with $U_i$ as jointly independent noises, which implies a very particular case of SCMs where there is no unmeasured confounders (i.e., a Markovian system). This is not the general definition of an SCM (as in Pearl, 2009) and should be clarified as an additional (and strong) assumption of the work.

**Strengths And Weaknesses:**

The paper addresses the important and relevant problem of identifying and estimating counterfactuals in temporal point processes.

However some assumptions are not clearly stated in the paper. The assumption that the $U_i$ variables are jointly independent noise is equivalent to the assumption that the system is Markovian (i.e., absence of unmeasured confounders between all pairs of variables). Moreover, the proposed model makes the assumption that $U_i$ is an additive noise.

Those assumptions are not part of the general definition of an SCM (as in Pearl, 2009), but are being used in the proposed sampling algorithm. They must be clearly stated as additional assumptions of the method and a their plausibility should be discussed.  Those are strong assumptions, most likely violated in many practical setting, so it is a critical limitation of the method. Further, the empirical study should address the robustness of the method under violation of those assumptions.

Without a clear statement of the underlying assumptions and a more detailed empirical study assessing robustness against violations of the assumptions, it is hard to judge the soundness and relevance of the proposed method.

---

> ### Author Response · Authors · 2022-07-27
> **Response to Reviewer 7bXC**
>
> We appreciate the reviewer's time and comments. We have updated the paper accordingly and, in what follows, we provide a point-by-point response.
>
> **[No unmeasured confounding / jointly independent noise variables]**
>
> In Section 2, our aim was to provide a brief introduction to SCMs. In lines 111-112 and footnote 3 of the revised submission, we have clarified that, in general, the noise variables in Eq. 3 may not be jointly independent if there are unmeasured confounders. Moreover, in lines 112-113, we have also clarified that, in our work, we assume there are no unmeasured confounders.
>
> In this context, we would like to emphasize that no unmeasured confounding is a very common assumption in the causal inference literature. In certain settings, there exist some solutions for the estimation of average treatment effects under unmeasured confounding (e.g., instrumental variables). However, we are unaware of any solution for the estimation of counterfactual distributions under unmeasured confounding, which we leave for future work.

---

> > ### Comment · Reviewer_7bXC · 2022-08-08
> > **Thank you for your response**
> >
> > **
> > _In this context, we would like to emphasize that no unmeasured confounding is a very common assumption in the causal inference literature. In certain settings, there exist some solutions for the estimation of average treatment effects under unmeasured confounding (e.g., instrumental variables). However, we are unaware of any solution for the estimation of counterfactual distributions under unmeasured confounding, which we leave for future work._
> > **
> >
> > No unmeasured confounding is not a common assumption in the causal inference theory. The entire causal framework proposed by Pearl and all recent advances in this field accepts the presence of unmeasured confounding. Note that the presence of unmeasured confounding is represented in a causal diagram by a bidirected edge. For a technical introduction to the field, refer to [Pearl, 2000] and [Bareinboim et al., 2022].  The fact that the authors are not at all familiar with these fundamental concepts in the field explains the several misconceptions in the paper.
> >
> > Moreover, counterfactual reasoning in the presence of unmeasured confounding is a well-studied problem. For example, [Shpitser and Pearl, 2008] provide methods to perform counterfactual reasoning in many (but not all) settings. Some of the conditions in that paper were relaxed in a more recent work by [Shpitser & Sherman, 2018]. The work by [Correa et al., 2021] claims to be complete for arbitrary nested counterfactual identification.
> >
> > **References**
> >
> > [Pearl, J. 2000] Judea Pearl (2000). Causality: Models, Reasoning, and Inference. Cambridge University Press. ISBN 978-0-521-77362-1. OL 15495862M. Wikidata Q108441215.
> >
> > [Bareinboim et al., 2022] Elias Bareinboim, Juan D. Correa, Duligur Ibeling, and Thomas Icard. 2022. On pearl’s hierarchy and the foundations of causal inference. In Probabilistic and Causal Inference: The Works of Judea Pearl, pp. 507-556.
> >
> > [Shpitser and Pearl, 2008] Shpitser, I. and Pearl, J., 2008. Complete identification methods for the causal hierarchy. Journal of Machine Learning Research, 9, pp.1941-1979.
> >
> > [Shpitser & Sherman, 2018] Shpitser, I. and Sherman, E., 2018, August. Identification of personalized effects associated with causal pathways. In Uncertainty in artificial intelligence: proceedings of the... conference. Conference on Uncertainty in Artificial Intelligence (Vol. 2018). NIH Public Access.
> >
> > [Correa et al., 2021] Juan D. Correa, Sanghack Lee, and Elias Bareinboim. "Nested counterfactual identification from arbitrary surrogate experiments." Advances in Neural Information Processing Systems 34 (2021): 6856-6867.

---

> > > ### Author Response · Authors · 2022-08-08
> > > **Re: Thank you for your response**
> > >
> > > Many thanks for taking the time and effort to engage with us.
> > >
> > > In our response, we did not mean to neglect the importance of unmeasured confounding in the field of causal inference and we do appreciate that you point us to recent advances in counterfactual reasoning in the presence of unmeasured confounding in your follow-up message.
> > >
> > > Following your original review, in the revised version of the paper, we did acknowledge that we assume no unmeasured confounders in our work. In this context, we would like to clarify that, in our revised paper, we just stated our assumption and we did not discuss how common this assumption is in the literature. More specifically, in lines 112-113, we wrote: "However, in our work, we will assume there are no unmeasured confounders and $U$ are jointly independent.".
> > >
> > > However, in what follows, we would like to respectfully disagree with the statement that **all** recent advances in the field of causal inference assume the presence of unmeasured confounding. Below, we provide a few examples of papers published in NeurIPS 2021 that explicitly assume no unmeasured confounding:
> > >
> > > - S. Sussex, A. Krause, C. Uhler, "Near-Optimal Multi-Perturbation Experimental Design for Causal Structure Learning", NeurIPS 2021. In page 1, the authors write: "Here we focus on the identification of DAGs that have no unobserved confounding variables."
> > >
> > > - I. Bica, D. Jarret, M. van der Schaar, "Invariant Causal Imitation Learning for Generalizable Policies", NeurIPS 2021. In page 2, the authors write: "We shall operate in the setting where there are no hidden confounders".
> > >
> > > - C. Cundy, A. Grover, S. Ermon, "BCDNets: Scalable Variational Approaches for Bayesian Causal Discovery", NeurIPS 2021. In page 6, the authors write: "Our approach assumes that no unobserved confounders are present".
> > >
> > > - V. Aglietti, N. Dhir, J. González, T. Damoulas, "Dynamic Causal Bayesian Optimization", NeurIPS 2021. In page 4, the authors write: "We make the following assumptions: [...] Absence of unobserved confounders".
> > >
> > > - S. Tsirtsis, A. De, Gomez-Rodriguez, "Counterfactual Explanations in Sequential Decision Making Under Uncertainty", NeurIPS 2021. In page 3, the authors write that, in their work, they consider a SCM with independent noise variables and no unobserved variables.
> > >
> > > - X. Chen, H. Sun, C. Ellington, E. Xing, L. Song, "Multi-task Learning of Order-Consistent Causal Graphs", NeurIPS 2021. In page 3, the authors write that, in their work, they consider a SCM with independent noise variables and no unobserved variables.
> > >
> > > In this context, we would also like to clarify that the Gumbel-Max SCM, as introduced by Oberst and D. Sontag, ICML 2019, also assumes no unmeasured confounding.
> > >
> > > That being said, in our first response, we would like to acknowledge that the use of the wording **very common** was a little unfortunate since what **very common** means is up to interpretation.

---

### Comment · Area_Chair_5KRE · 2022-08-08
**Please respond to author response**

Dear reviewers,

Thank you for reviewing this paper. Could you respond to the author feedback, or at least acknowledge that you have read the response (thanks to reviewer E24B for having done it)? Please indicate whether the author response addresses your concerns.

Thanks,
AC

---

### Meta-Review · Area_Chair_5KRE · 2022-08-29

**Recommendation:** Accept
**Confidence:** Less certain

**Metareview:**

This paper proposed a simple yet sensible method to answer counterfactual questions for temporal point processes. Specifically, the authors focus on the counterfactual question of whether a historical event would have happened if the corresponding intensity had been changed. Reviewers agree that the idea is clearly presented and theoretically plausible, with interesting and important epidemiological applications. Please also pay attention to the reviewers' concerns about the paper title and the presentation of some specific parts of the paper.

**Award:**

No

---

### Decision · Program_Chairs · 2022-09-14

Accept